# Bacteriuria in Cystocentesis Samples from Cats in the United Kingdom: Prevalence, Bacterial Isolates, and Antimicrobial Susceptibilities

**DOI:** 10.3390/ani12233384

**Published:** 2022-12-01

**Authors:** Clarisse D’Août, Samantha S. Taylor, Stefania Gelendi, Carl Atkinson, Pieter Defauw

**Affiliations:** 1Internal Medicine Department, Lumbry Park Veterinary Specialists, Selborne Road, Alton GU34 3HL, UK; 2Royal (Dick) School of Veterinary Studies, University of Edinburgh, Easter Bush Campus, Midlothian EH25 6RG, UK; 3Axiom Veterinary Laboratories Ltd., Manor House, Brunel Rd, Newton Abbott TQ12 4PB, UK

**Keywords:** urinary tract infections, feline lower urinary tract disease, urine culture, antimicrobial resistance

## Abstract

**Simple Summary:**

Bacterial urinary tract infections are one of the most important reasons for the use of antimicrobials in veterinary medicine. Prevalence and resistance can vary greatly with geographical location and are poorly reported for the UK. This study investigated the prevalence of positive cultures in a population of cats with urine samples collected by cystocentesis and submitted to a laboratory in the UK, as well as the isolated bacterial species and their antimicrobial susceptibility and resistance to commonly used antibiotics.

**Abstract:**

Bacterial urinary tract infections (UTIs) have historically been reported to be uncommon in cats; however, recent studies showed a higher prevalence. Bacterial UTIs are one of the most common reasons for the use of antimicrobial drugs in veterinary medicine. Our aim was to investigate the prevalence of positive cultures in urine samples submitted to a UK laboratory for testing, as well as prevalence of bacterial species and their antimicrobial susceptibility to commonly used antibiotics. This was a retrospective analysis of positive cultures from feline urine samples collected by cystocentesis submitted over 14 months (January 2018–February 2019). A total of 2712 samples were reviewed, of which 425 documented a positive culture (15.7%) with a total of 444 bacterial isolates. *E. coli* (43.7%), other *Enterobacterales* (26.4%), *Enterococcus* species (14.9%) and *Staphylococcus* species (9.2%) were the most commonly isolated bacteria. *E. coli* most commonly showed resistance to cephalexin (20.7%) and amoxicillin (16.7%). Resistance was most commonly seen against amoxicillin (64.1%) and cephalexin (52.2%) in *Enterobacterales*. *Enterococcus* species most commonly showed resistance to trimethoprim/sulfamethoxazole (94.3%). *Staphylococcus* species most commonly showed resistance to amoxicillin (20%). This study showed significant resistance of bacteria found in feline urine samples in the UK to frequently used antibiotics.

## 1. Introduction

Bacterial UTIs have historically been reported to be uncommon in cats, with a prevalence of less than 3% among cats with signs of feline lower urinary tract disease (FLUTD) [1]. However, more recent studies have reported a much higher prevalence of 8–19% in cats with FLUTD in Europe [2,3,4]. Affected cats typically have concurrent diseases, commonly chronic kidney disease (CKD), diabetes mellitus, or hyperthyroidism [5,6,7,8]. Increasing age and female sex are identified as risk factors for UTIs [7]. Subclinical bacteriuria (SB) has also been reported in cats with a highly variable prevalence (1–29%) [7,9,10,11,12]. However, FLUTD is one of the most common reasons for the use of antimicrobial drugs in veterinary medicine [13,14,15]. To prevent further development of antimicrobial resistance, selection of antimicrobial agents should be based on urine culture results and in vitro susceptibility testing in cats, and antimicrobial agents with an appropriate but narrow spectrum of activity should be used. Previous studies have shown that *Escherichia coli* (*E. coli*), *Streptococcus* species, *Enterococcus* species, *Staphylococcus* species, and *Micrococcaceae* were the most commonly isolated bacteria in feline UTIs [6,7,16,17,18,19]. Fonseca and others (2021) [17] documented the antimicrobial susceptibility of 171 positive urine cultures obtained from cats in the UK and concluded that a third of *E. coli* isolates were resistant to ampicillin, but that resistance was much lower among *Enterococcus* species. They also reported low resistance to trimethoprim/sulfamethoxazole (TMPS) for all uropathogens. Dorsch and others (2015) [16] documented 330 culture-positive urine specimens (375 isolates) in Germany, and reported that *Streptococcus* and *Enterococcus* isolates were resistant to a significantly higher number of antimicrobial agents than *E. coli* and *Staphylococcus* species. Lund and others (2014) [20] documented the antimicrobial susceptibility of 82 bacterial isolates in Norway; *E. coli, Staphylococcus*, *Enterococcus*, and *Streptococcus* species were most frequently isolated, and although several tendencies towards increasing antimicrobial resistance were detected among the isolates included, the species of bacteria isolated and their patterns of antimicrobial resistance were in concurrence with the existing literature. However, prevalence and resistance can vary significantly with geographical location, and with the exception of the study by Fonseca and others (2021) [17], the epidemiology of feline UTIs in the United Kingdom (UK) is underreported and may differ from that reported in other countries. The aim of the present study was to identify prevalence of positive cultures in a population of cats with urine samples collected by cystocentesis and submitted to a laboratory in the UK and analyse their antimicrobial susceptibilities.

## 2. Materials and Methods

### 2.1. Animals and Samples

Urinalysis results from feline urine samples submitted to and analysed by Axiom Veterinary Laboratories and Finn Pathologists over a period of 14 months (January 2018–February 2019) were reviewed. To be included, urine samples had to be obtained by cystocentesis, and signalment information recorded (breed, age, sex, and neutering status). When multiple urine samples had been submitted, only the first result was included in the study. Presence of lower urinary signs defined as dysuria, stranguria, pollakiuria or haematuria was recorded when available. This work involved the use of non-experimental animals only (owned and unowned), and followed internationally recognised high standards (‘best practice’) of individual veterinary clinical patient care. This study was approved by the CVS Ethics committee (approval number CVS-2020-02).

### 2.2. Urine Culture

For semi-quantitative aerobic culture of urine samples, specimens were well mixed before inoculating 10µL of specimen onto Cystine-lactose-electrolyte-deficient (CLED) agar (Thermo Fisher Diagnostics, Waltham, MA, USA), Columbia agar with 5% sheep blood and, for antimicrobial sensitivity testing, Mueller–Hinton agar (all supplied by Thermo Fisher Diagnostics, Waltham, MA, USA), respectively (see subsequent paragraph). Streak dilution techniques were performed on CLED and Columbia agars, whereas a ‘lawn’ spread technique was performed on Mueller–Hinton agar. All plates were incubated at 37 °C overnight and examined for growth. The plates were examined after incubation and results recorded. Semi-quantitative grading of each different population of bacteria was performed using scant (<10^4^ colony forming unit (CFU)/mL), light (10^4^ CFU/mL), moderate (10^5^ CFU/mL) and profuse (10^6^ CFU/mL) growth distinctions. Identification of bacterial isolates was based on a multitude of factors including phenotypical (macroscopic as well as microscopic) appearance, basic biochemistry (indole, oxidase, catalase reactions as required) or further detailed typing. If further typing was required, the most common methods involved Analytical Profile Index (API)(Biomerieux, Marcy-l’Etoile, France, performed in the lab following manufacturer’s guidelines) or Matrix-Assisted Laser Desorption/Ionisation Time-of-Flight (MALDI-ToF) (organisms expedited to Laboklin Laboratory, Bad Kissingen, Germany). When criteria for *E. coli* or *Proteus* species were not met, but the isolates were identified as *Enterobacterales*, they were classified as ‘other *Enterobacterales*’. This group included *Enterobacter* species, lactose fermenting coliforms, and non-lactose fermenting coliforms.

### 2.3. Antimicrobial Susceptibility

Kirby–Bauer disc diffusion technique was used, placing antibiotic discs (supplied by Thermo Fisher Diagnostics, Waltham, MA, USA) on the Mueller–Hinton agar. Antimicrobial agents used included a ‘Urine panel’ consisting of amoxicillin, amoxicillin/clavulanic acid (AMC), cephalexin, pradofloxacin, marbofloxacin, cefovecin, and TMPS. Occasionally, and usually dependent upon the resistance pattern of the isolated organisms to the ‘Urine panel’ or if specifically requested by veterinarians, an ‘Extended panel’ was performed which also included ceftazidime, cefuroxime, tobramycin, gentamicin, amikacin, ticarcillin/clavulanic acid, piperacillin/tazobactam and nitrofurantoin. Furthermore, individual agents could also be requested on an ad hoc basis by clients but would not be tested as standard. Antimicrobial susceptibility patterns were recorded for all agents and all identified bacteria. The isolates were described as ‘Sensitive’, ‘Intermediate’ or ‘Resistant’ as determined by inhibition zone diameter. Zone size breakpoints, as published by the Clinical and Laboratory Standard Institute [21,22], Eucast and those supplied by disc suppliers were used as a basis for this interpretation. Where no breakpoint data were available, extrapolation from similar organisms/antimicrobials was performed. For the purpose of this study, isolates described as ‘Intermediate’ were classified as ‘Resistant’.

### 2.4. Statistical Analysis

The results are presented as frequencies of occurrence, expressed in percentages.

## 3. Results

### 3.1. Animals and Samples

A total of 2712 cystocentesis samples were analysed, of which 425 had a positive culture (15.7%). Three cats had two positive culture results from two different samples, and only the first result was included in this study as there was a concern the second sample was requested following antimicrobial therapy (not specified). A total of 162 male cats and 262 female cats were included. The neutering status of several cats was retrospectively found to not have been updated on computer records, and due to general data protection regulation limitations could not be verified for all cats included in this study. It was therefore not included in analysis. The mean age was 11 years 7 months (SD ± 4.864 years; median 12.3 years, range from 5 months to 21 years 2 months).

Lower urinary signs were reported in 25.1% of the cases. Presence or absence of lower urinary signs was not specified in the other cats. Out of the 425 culture-positive specimens, 406 (95.5%) resulted in growth of a single bacterial isolate, while in the remaining 19 cultures (4.5%), two different isolates were detected. A total of 444 bacterial isolates were obtained.

### 3.2. Bacterial Isolate Identification

*E. coli* (43.7%), other *Enterobacterales* (26.4%), *Enterococcus* species (14.9%) and *Staphylococcus* species (9.2%) were the most commonly isolated bacteria (Figure 1). Non-Beta haemolytic *Streptococcus* species, *Pseudomonas* species, *Proteus* species, *beta-haemolytic Streptococcus* species and *Pasteurella* species accounted for 3.2%, 1.4%, 0.9%, 0.45%, and 0.2% of all isolates, respectively.

### 3.3. Antimicrobial Susceptibility

Antimicrobial susceptibilities and resistance were tested for all bacterial isolates (Table 1). Additional figures are provided for *E. coli* (Figure 2), other *Enterobacterales* (Figure 3), *Enterococcus* species (Figure 4), and *Staphylococcus* species (Figure 5).

## 4. Discussion

In the present study, bacterial prevalence in a population of cats with positive urine cultures from a large number of samples collected by cystocentesis and submitted to a laboratory in the UK, as well as their antimicrobial susceptibilities, were investigated. With more than 400 culture-positive urine samples collected by cystocentesis only, this study included a large data set, with samples obtained from both primary care practices and referral centres. To the authors’ knowledge, this is the largest study describing the prevalence, bacterial isolates, and antimicrobial susceptibilities for bacteriuria in a population of cats with positive urine culture specifically in the UK. Fonseca and others (2021) investigated the antimicrobial resistance of uropathogens of cats and dogs in the UK, but the method of urine collection was not specified, the time period studied was several years prior to the data presented in this study, and a smaller number of cats were included [17]. Marques and others (2016) published a European multicentre study on antimicrobial resistance in bacteria isolated from feline and canine UTIs [23], but no differentiation between species was available for the assessment of antimicrobial resistance, and the results obtained therefore cannot be applied to the feline population in the UK.

Bacteriuria was documented in 15.7% of the submitted samples. Although this seems in concordance with the existing literature (1–29%) [7,9,10,11,12,17] and slightly higher than in the study performed by Fonseca and others (2021) (10%) [17], prevalence of positive culture results in this population is dependent on a poorly characterised population and therefore does not represent prevalence of bacteriuria in cats in the UK.

*E. coli* (43.7%), other *Enterobacterales* (26.4%), *Enterococcus* species (14.9%) and *Staphylococcus* species (9.2%) were the most commonly isolated bacteria in the present study. The prevalence of *E. coli* in feline UTI has previously been reported to be 37–71% [4,6,8,16,17,18,20,23,24] and the prevalence obtained in this study correlates with these data. Other *Enterobacterales* was the second most commonly isolated bacterial group in this study, and this regrouping was established to include unclassified *Enterobacteriaceae* while respecting the bacterial taxonomy [25]. To the authors’ knowledge, no previous studies have included this bacterial group in their results, which makes comparison challenging. Prevalence of *Enterococcus* species (6.6–27%) and *Staphylococcus* species (7.8–22.9%) were similar in previous studies [4,6,16,17,18,20,23,24]. Fonseca and others (2021) had a higher prevalence of *Enterococcus* species (23.2%) than the one observed in the present study (14.9%) [17].

Antimicrobial resistance findings reported in this study only relate to this population of isolates, and cannot be automatically applied to other isolates.

*E. coli* most commonly showed resistance to amoxicillin (16.7%) and cephalexin (20.7%); however, this bacteria’s susceptibility to these antimicrobials agents in the present study was generally high and much higher than in a previous study performed in Norway where *E. coli* resistance for these agents was 64% and 60%, respectively [20]. This is an encouraging finding given *E. coli* was the most common isolate. Fonseca and others (2021) showed much higher resistance of *E. coli* to AMC (19.2%), TMPS (8.2%) and cefovecin (26.8%) [17] compared with the present study where resistance was 5.7%, 0.5% and 2.6%, respectively. Our larger dataset (199 *E. Coli* isolates in the present study vs. 94 isolates in the study of Fonseca and others) could explain this difference.

It is challenging to compare the antimicrobial susceptibility testing results for other *Enterobacterales* with other studies, as this bacterial group has not been studied per se previously. However, the present study showed high resistance patterns for penicillins (39–64%), and Lund and others (2014) also showed a high level of resistance to penicillins in gram-negative rods (60–90%) [20]; they suggested that this group of bacteria may not be effectively treated by these commonly selected antibiotics. Although the results obtained for other *Enterobacterales* would support this, it is important to note that the *E. coli* resistance to penicillins were low in the present study.

*Enterococcus* species showed a high resistance to TMPS. *Enterococcus* species are frequently found to be resistant to several antibiotics [5,6,20,26,27], and this was also the case in this study, although this bacterium was seldom resistant to amoxicillin (2.9%), AMC (0.0%), and nitrofurantoin (0.0%). Intrinsic resistance against cephalosporins is expected in this prevalent group of bacteria [18] and the use of cephalosporins in the context of *Enterococcus* species (prevalence of 14.9%) is therefore problematic. It has indeed been reported that a large proportion of the total use of cefovecin in the UK (13.7%) is for treatment of lower urinary tract diseases, and in most cases (56%) this drug is prescribed due to an inability to orally medicate the cat [28]. Furthermore, a recent multicentre study of antimicrobial prescriptions for cats diagnosed with bacterial urinary tract disease in the USA and Canada concluded that cefovecin was the most commonly prescribed antibiotic, with a percentage of use varying between 27 and 61% depending on the type of urinary tract disease diagnosed [13]. This inappropriate use could not only be ineffective but promote further antimicrobial resistance. However, Weese and others (2021) also identified an increased consistency with the International Society for Companion Animals Infectious Diseases (ISCAID) guidelines over the 3-year study period, with less use of third generation cephalosporins and fluoroquinolones by study completion [13]. Although these changes were of relatively low magnitude, they are encouraging indicators.

What differs markedly between the present study and the previous literature is the *Enterococcus* species resistance to TMPS, which was much higher in this study (94.3%) than in a previous study performed in Norway (15%) [20]. Unfortunately, resistance of *Enterococcus* species to TMPS was not evaluated by Fonseca and others (2021) in the UK [17]. The differences in resistance for *Enterococcus* species to TMPS between the UK and Norway could reflect a geographical variation within the *Enterococcus* species population itself, or relate to a different use of antimicrobial agents between countries. However, nearly all tested *Enterococcus* species were susceptible to amoxicillin in the present study, which is positive considering this antimicrobial agent is acceptable in the treatment of uncomplicated UTI by the antimicrobial guidelines working group of the ISCAID [29,30] and is frequently prescribed in veterinary medicine in the UK.

In this study, *Staphylococcus* species did generally not represent a challenge with regard to antibiotic selection, although 20% of the cultured isolates was resistant to amoxicillin. The patterns of antimicrobial resistance were, in general, in concurrence with the existing literature [16,20,31,32].

This study has several limitations. Firstly, prevalence of positive culture results in this population is dependent on a poorly characterized population and therefore does not fully represent the prevalence of bacteriuria in cats in the UK. Secondly, due to the retrospective nature, recording of clinical information is not consistent. Despite the large number of cystocentesis samples obtained, information relating to clinical signs observed (especially presence or absence of lower urinary signs), potential previous FLUTD, comorbidities, and previous or ongoing antimicrobial treatment were not reported for most cases. Therefore, the prevalence of lower urinary tract signs (25.1%) observed in this study is, based on the available information, highly likely to be an underestimation. It is likely that many cats had lower urinary tract signs that were not recorded on sample submission forms, and therefore we cannot conclude the remaining cats (74.9%) had SB, limiting the value of the reported data. Further research studying bacterial isolates and antimicrobial susceptibilities in cats with confirmed lower urinary tract signs and/or comparing results obtained from cats with confirmed lower urinary tract signs with results from cats with confirmed SB would be welcome. Urinalysis results, including culture and sensitivity, should be interpreted with clinical signs, as current ISCAID recommendations are not to treat SB in most cases [29,30]. The results of the present study represent findings in a population of cats seen in both primary care and referral hospitals and potential differences between these two populations could not be assessed. It is highly probable that prior or ongoing antimicrobial treatment had an impact on the bacterial species isolated and on the number of antimicrobial agents that isolates were resistant to, but this information was not consistently available. The prevalence of sex was not analysed nor compared with a control population because the proportion of entire and neutered male and female was likely incorrect. The lack of consistently available MALDI-ToF machine on site meant further identification of some isolates was not possible. Measurement of minimal inhibitory concentration was not available, and this could have provided additional information for the interpretation of antimicrobial susceptibility testing results, especially as this could influence the decision to choose a certain antimicrobial agent in practice.

Some of the limitations encountered in this study are inherent to all studies based on laboratory analyses (e.g., external urinalysis requested due to lack of clinical response to previous antimicrobial treatment). The authors would therefore like to suggest the use of Flexicult^®^ Vet in future studies to explore or avoid these limitations. Flexicult^®^ Vet is a culture-based point of care test for detection, identification, and antimicrobial susceptibilities testing of bacterial uro-pathogens in veterinary practice [33]. Although bacterial identification has been reported to be operator-dependent in previous studies [33,34] and adequate clinical staff training would therefore be needed, including this technique in further studies would not only involve studying populations that are more representative of the feline population seen in first opinion practices, but also hopefully discourage empirical use of cefovecin and encourage adherence to treatment guidelines.

## 5. Conclusions

*E. coli*, other *Enterobacterales*, *Enterococcus* species, and *Staphylococcus* species were the most commonly isolated bacteria in this study. Resistance to frequently used antibiotics was documented, especially in the case of *Enterococcus* species, which was much less susceptible to TMPS compared to other studies. In contrast, resistance was limited in the case of *E. coli*. The pattern of antimicrobial susceptibility in the present study was otherwise in agreement with the existing literature. However, the lack of recording of clinical signs limits interpretation. More research studying a population with known presence or absence of lower urinary signs, comorbidities and previous or ongoing antimicrobial treatment would be welcome.

## Figures and Tables

**Figure 1 animals-12-03384-f001:**
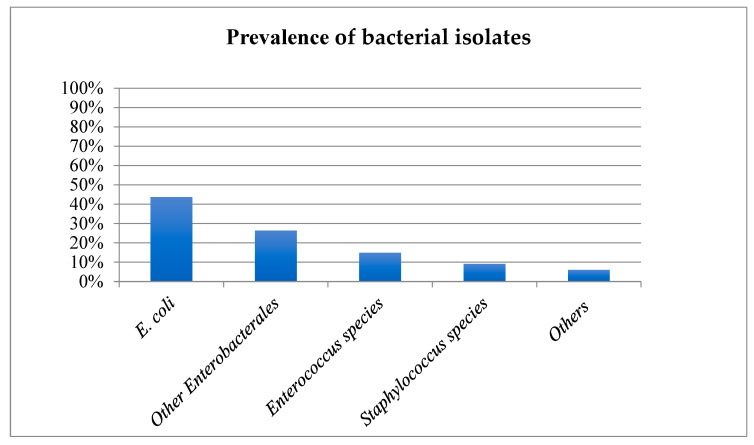
Prevalence of bacterial species isolated from culture-positive feline urine samples. X axis: Bacteria cultures, y axis: Percentage of isolates.

**Figure 2 animals-12-03384-f002:**
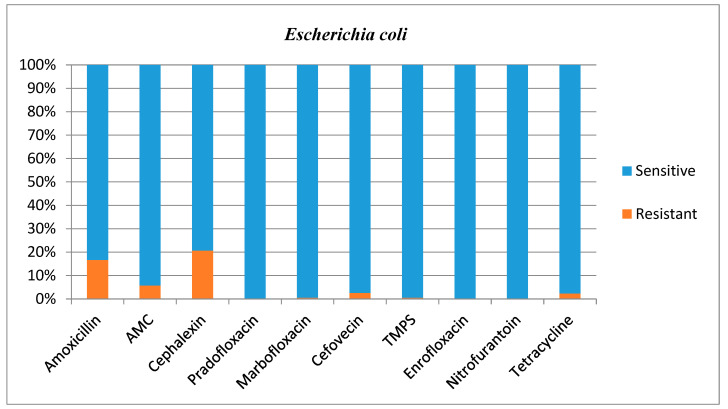
Antimicrobial susceptibility testing results, reporting sensitivities and resistance of *E. coli.* AMC **=** amoxicillin/clavulanic acid, TMPS = trimethoprim/sulfamethoxazole.

**Figure 3 animals-12-03384-f003:**
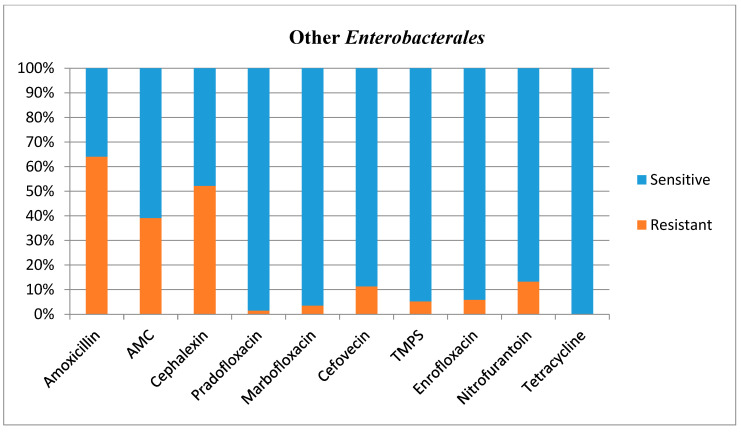
Antimicrobial susceptibility testing results, reporting sensitivities and resistance of Other *Enterobacterales.* AMC **=** amoxicillin/clavulanic acid, TMPS = trimethoprim/sulfamethoxazole.

**Figure 4 animals-12-03384-f004:**
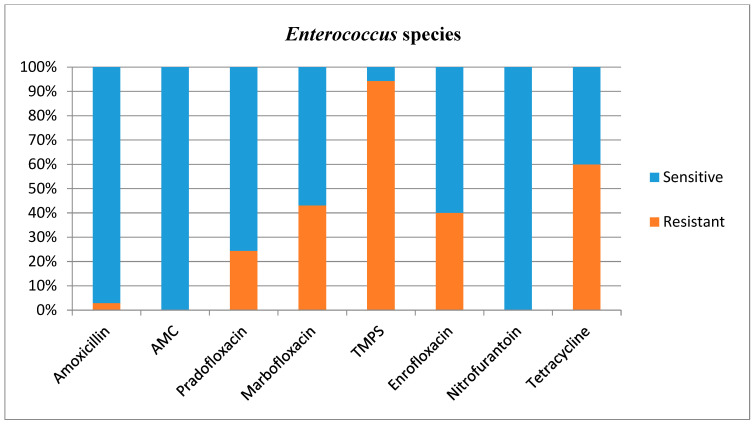
Antimicrobial susceptibility testing results, reporting sensitivities and resistance of *Enterococcus* species. AMC = amoxicillin/clavulanic acid, TMPS = trimethoprim/sulfamethoxazole.

**Figure 5 animals-12-03384-f005:**
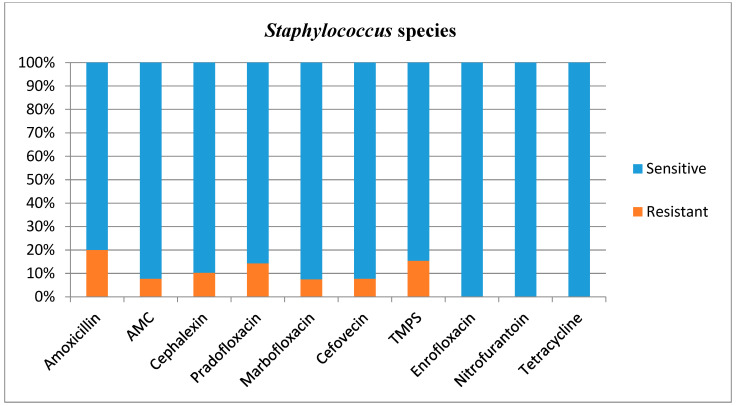
Antimicrobial susceptibility testing results, reporting sensitivities and resistance of *Staphylococcus* species. AMC = amoxicillin/clavulanic acid, TMPS = trimethoprim/sulfamethoxazole.

**Table 1 animals-12-03384-t001:** Antimicrobial susceptibility testing results, reported resistance (in percentage and total numbers) of bacterial isolates from positive urine cultures collected by cystocentesis. *E. Coli = Escherichia coli*.

	Amoxicillin	Amoxicilln/Clavulanic Acid	Ticarcillin	Ticarcillin/Amoxiclav	Piperacillin	Piperacillin/Tazobactam	Cephalexin	Cefovecin	Ceftazidime	Cefuroxime	Pradofloxacin	Enrofloxacin	Marbofloxacin	Trimethoprim/Sulfamethoxazole	Nitrofurantoin	Tetracycline	Gentamicin	Tobramycin	Erythromycin	Polymyxin B	Chloramphenicol
*E. Coli* (199/455)	16.7%(16/96)	5.7% (11/193)					20.7% (40/193)	2.6% (5/192)			0.0% (0/112)	0.0% (0/97)	0.5% (1/193)	0.5% (1/193)	0.0% (0/44)	2.3% (1/43)					
Other *Enterobacterales* (119/455)	64.1% (41/64)	39.1% (45/115)	100% (1/1)	100% (1/1)	0.0% (0/1)	0.0% (0/1)	52.2% (60/115)	11.3% (13/115)	0.0% (0/1)		1.5% (1/66)	5.9% (3/51)	3.5% (4/115)	5.2% (6/115)	13.3% (4/30)	0.0% (0/1)	0.0% (0/1)	0.0% (0/1)			
*Enterococcus species* (70/455)	2.9% (1/35)	0.0% (0/65)				100% (1/1)					24.4% (10/41)	40.0% (12/30)	43.1% (28/65)	94.3% (33/35)	0.0% (0/11)	60.0% (9/15)	100% (1/1)	100% (1/1)	0.0% (0/1)	100% (1/1)	
*Pseudomonas species* (6/455)	100% (3/3)	100% (6/6)	100% (1/1)	100% (2/2)	0.0% (0/1)	0.0% (0/2)	100% (6/6)	100% (6/6)	0.0% (0/2)	0.0% (0/1)	0.0% (0/3)	33.3% (1/3)	16.7% (1/6)	100% (6/6)	100% (2/2)		0.0% (0/2)	0.0% (0/2)			
*Beta-haemolytic Streptococcus* (2/455)	0.0% (0/2)	0.0% (0/2)					0.0% (0/2)	0.0% (0/2)			0.0% (0/2)		0.0% (0/2)	0.0% (0/2)							
*Streptococcus species* (14/455)	16.7% (1/6)	0.0% (0/15)					40.0% (6/15)	26.7% (4/15)			25.0% (2/8)	44.4% (4/9)	33.3% (5/15)	13.3% (2/15)	33.3% (1/3)	25.0% (1/4)					
*Staphylococcus species* (41/455)	20.0% (4/20)	7.7% (3/39)					10.3% (4/39)	7.7% (3/39)			14.3% (3/21)	0.0% (0/20)	7.5% (3/40)	15.4% (6/39)	0.0% (0/7)	0.0% (0/12)	100% (2/2)		0.0% (0/1)		0.0% (0/2)
*Pasteurella species* (1/455)	0.0% (0/1)	0.0% (0/1)					0.0% (0/1)	0.0% (0/1)			0.0% (0/1)		0.0% (0/1)	0.0% (0/1)							
*Proteus species* (4/455)		0.0% (0/4)					25.0% (1/4)	0.0% (0/4)				0.0% (0/4)	0.0% (0/4)	25.0% (1/4)	25.0% (1/4)	25.0% (1/4)					

## Data Availability

The data presented in this study are available in the article. Full dataset and further information are available upon request from the corresponding author.

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
