# Peer review of "Bacteriuria in Cystocentesis Samples from Cats in the United Kingdom: Prevalence, Bacterial Isolates, and Antimicrobial Susceptibilities"

_animals, 2022, doi:10.3390/ani12233384_

Round 1
Reviewer 1 Report
This is a very well-written article. The data provided is interesting but must be interpreted with great caution given the limited characterisation of the sampled population. Further comments are provided in the attached document.

Reviewer 2 Report
D’Aout et al. presented a study on the prevalence of bacteria and their antimicrobial resistance in feline UTIs. The paper is well-written, and the results are presented clearly. I have four (relatively minor) comments, listed in order of their appearance:
1) Page 3, line 127. What does it mean that three cats had two positive results? That multiple bacteria were detected or, as you mentioned in the M&M, more urine samples were submitted?
2) Page 3, lines 142-143. If available, please include, which Enterococcus and Staphylococcus species were the most common. Additionally, it would be very interesting to see if you have any statistics on MRSP/MRSA bacteria.
3) Page 4 & 5, Figure 1, Table 1. In the caption, explain E. coli abbreviation.
4) Page 8-10, lines 227-236 & 286-292. The authors mentioned a problematic empirical use of third-line antibiotics in UTIs, poor adherence to the guidelines, and a need for first-line antibiotics use. Here, I suggest additionally mentioning (and discussing) Flexicult Vet, a point-of-care test for bacterial UTI, which offers basic bacterial identification and AST (mostly for first-line antibiotics). Two useful references for Flexicult Vet: introduction (Guardabassi, 2015, doi: 10.1186/s13028-015-0165-4) and practical challenges (Cugmas, 2021, doi: 10.3390/antibiotics10101160).
Reviewer 3 Report
The manuscript entitled “Bacteriuria in cystocentesis samples from cats in the United Kingdom: prevalence, bacterial isolates, and antimicrobial susceptibilities “is very well written and discussed an important issue of antimicrobial resistance. The article is simply written and easily understandable for the readers. In introduction, the issue is properly discussed. Materials and methodological section elaborated all techniques used in the article in simplified manner. Results are clearly explained and supported by the graphical representation. Findings of the study are comprehensively discussed with updated references of previous studies. In my opinion, the study can be extended by further studying the isolated cultures using molecular techniques that will help in understanding the mechanism of antimicrobial resistance in different isolates of bacteria. Best regards.
Round 2
Reviewer 1 Report
See comments to editor.